# An Improved STL-LSTM Model for Daily Bus Passenger Flow Prediction during the COVID-19 Pandemic

**DOI:** 10.3390/s21175950

**Published:** 2021-09-04

**Authors:** Feng Jiao, Lei Huang, Rongjia Song, Haifeng Huang

**Affiliations:** 1Department of Information Management, School of Economics and Management, Beijing Jiaotong University, Beijing 100044, China; 15120625@bjtu.edu.cn (F.J.); rjsong@bjtu.edu.cn (R.S.); 17113151@bjtu.edu.cn (H.H.); 2Department of Decision Sciences and Information Management, Faculty of Economics and Business, KU Leuven, 3000 Leuven, Belgium

**Keywords:** daily bus passenger flow prediction, hybrid model, deep learning, COVID-19

## Abstract

The COVID-19 pandemic is a significant public health problem globally, which causes difficulty and trouble for both people’s travel and public transport companies’ management. Improving the accuracy of bus passenger flow prediction during COVID-19 can help these companies make better decisions on operation scheduling and is of great significance to epidemic prevention and early warnings. This research proposes an improved STL-LSTM model (ISTL-LSTM), which combines seasonal-trend decomposition procedure based on locally weighted regression (STL), multiple features, and three long short-term memory (LSTM) neural networks. Specifically, the proposed ISTL-LSTM method consists of four procedures. Firstly, the original time series is decomposed into trend series, seasonality series, and residual series through implementing STL. Then, each sub-series is concatenated with new features. In addition, each fused sub-series is predicted by different LSTM models separately. Lastly, predicting values generated from LSTM models are combined in a final prediction value. In the case study, the prediction of daily bus passenger flow in Beijing during the pandemic is selected as the research object. The results show that the ISTL-LSTM model could perform well and predict at least 15% more accurately compared with single models and a hybrid model. This research fills the gap of bus passenger flow prediction under the influence of the COVID-19 pandemic and provides helpful references for studies on passenger flow prediction.

## 1. Introduction

Since COVID-19 appeared in Wuhan, China, in December 2019, it has spread rapidly around the world. On 11 March 2020, WHO stated that the current COVID-19 outbreak should be called a global pandemic. Globally, until 1 September 2020, there had been 25,685,207 confirmed cases of COVID-19, including 899,488 deaths [1]. This global pandemic has brought life and work troubles to people all over the world. In terms of passenger transport, buses, subways, and airplanes are fast ways for the pandemic to spread due to their confined space. Therefore, whether people choose the above ways to travel depends largely on the actual situation of the local pandemic.

Take the bus as an example: the COVID-19 pandemic has dramatically affected the everyday travel of passengers; many passengers give up taking the bus to avoid touching each other in the limited space on the bus, while those who choose to take buses are at high risk of getting infected at any time. Besides, the pandemic has also created many issues for public transport companies, such as Beijing Public Transportation Corporation (BPTC), on scheduling and management: for example, the imbalance between capacity and demand caused by a rapid decline in passenger flow, low operational efficiency, and the maximum number limitation of passengers on each bus.

Meanwhile, with the rapid development and continuous improvement of urban public transport information technology, public traffic management information systems have produced much bus passenger travel data. If prediction approaches are employed using these data to scientifically compute the variation trend of passenger flows in the future, it will help public transport companies gain the following advantages:Optimizing operation scheduling of buses and staff.Improving operational efficiency.Reducing operating costs.Preventing and controlling the spread of the epidemic in buses.Promoting intelligent management information systems.Enhancing emergency response capacity and dealing with emergencies promptly.Improving service levels and competitiveness.

Furthermore, it will also benefit passengers, such as by reducing the risk of being infected, reducing the time cost of waiting for a bus, and improving their comfort. Therefore, it is significant for both passengers and companies to establish a model to predict the daily bus passenger flow based on COVID-19 and other data comprehensively, accurately, and scientifically.

This research aims to predict the daily bus passenger flow during an epidemic or pandemic disease. The significant contributions of this research can be briefly summarized into three aspects:This research analyzes the impact of the COVID-19 pandemic on bus passenger flow and introduces COVID-19 data into bus passenger flow prediction. The experimental results of the case study prove that in predicting bus passenger flow during the COVID-19 pandemic, models with holiday, weather, and COVID-19 data features perform better than models without any such features. Even some single models with these features can perform better than hybrid models without them.In this research, the seasonal-trend decomposition procedure based on locally weighted regression (STL) is applied in bus passenger flow prediction. There are three components for the passenger flow time series of the bus: a trend series, a seasonal series, and a residual series. The experimental results of the case study indicate that the models with STL have better performance than the models without STL in the prediction of bus passenger flow under the influence of the COVID-19 pandemic.An improved STL-LSTM model (ISTL-LSTM) is proposed for passenger flow prediction. It consists of an STL model, many related features of passenger flow, and three long short-term memory (LSTM) models. In the case study, the experiment results show that the ISTL-LSTM model achieves higher accuracy (at least 15%) than other models regarding bus passenger flow during the COVID-19 pandemic. Therefore, it can provide public transport companies with high-accuracy passenger flow prediction to help make more reasonable arrangements for the COVID-19 pandemic.

## 2. Related Work

The prediction of passenger flow is very similar to that of traffic flow [2,3,4,5]. Therefore, it is also possible to refer to the literature related to the prediction of traffic flow. Methodologically, different scholars have different classifications of the passenger/traffic flow prediction approaches. Some scholars divided prediction methods into parametric approaches and non-parametric approaches [6,7,8,9]. Li et al. divided them into three categories: linear models, nonlinear models, and combined models [10]. In comparison, Luo et al. divided them into three categories: linear approach, nonlinear approach, and hybrid approach [11]. Zhou et al. divided them into four categories: parametric models, non-parametric models, hybrid models, and deep learning models [12]. Liu et al. divided them into traditional classical algorithms, regressive models, machine learning-based models, and hybrid models [13]. In this research, based on the classification method of Zhou et al. [12], passenger prediction models are separated into the following four categories: parametric models, non-parametric models, deep learning methods, and hybrid models.

Parametric models have the advantages of being easy to implement and having high computational efficiency, which can be applied to some specific traffic conditions. Some commonly used models are historical average, autoregressive integrated moving average (ARIMA) [14], Kalman filter (KF) [15], and exponential smoothing (ES) [16]. However, these models are less effective when dealing with complex passenger flow data.

Without making assumptions, non-parametric models can freely learn any function from the training data and fit well with the training data. Widely used non-parametric models include: artificial neural network (ANN) [17,18,19,20], k-nearest neighbor (KNN) [21,22,23], and support vector regression (SVR) [2,24] models. Compared with parametric models, non-parametric models usually perform better when dealing with large data sizes. Due to the fitting ability of nonlinear modeling, non-parametric models can deal with the fluctuation of passenger flow time series. Nevertheless, they have the drawbacks of over-fitting and calculation complexity.

In addition to parametric and non-parametric models, many recent studies have employed deep learning methods in passenger flow prediction, such as deep belief networks (DBNs) [25], LSTM [4,5], stacked autoencoders (SAE) [13], and convolutional neural networks (CNNs) [4,26,27,28]. Liu et al. [4] proposed an end-to-end deep learning architecture for short-term metro passenger flow prediction. Furthermore, Hao et al. [5] presented a sequence-to-sequence learning model embedded with an attention mechanism to predict short-term urban metro passenger flow. Liu et al. [13] proposed a multi-stage deep learning model to predict hourly passenger flow. Du et al. [27] developed a dynamic transition convolutional neural network for precise traffic prediction. According to these research results, it can be concluded that deep learning models usually have higher prediction accuracy than parametric models and non-parametric models. The reason is that complex neural networks have a universal fitting capability, and they can fit any nonlinear function theoretically. However, deep learning models have high computational complexity and require many resources and training time [29]. Besides, they are often regarded as black boxes, and the results lack interpretability [29].

Hybrid models integrate several approaches to overcome the shortcomings of single models and take full advantage of different models. Hybrid models can consider both linear and nonlinear characteristics of passenger flow and improve the accuracy of passenger flow prediction. Therefore, hybrid models have gradually become the research focus; for example, Li et al. [10] utilized a seasonal autoregressive integrated moving average model (SARIMA) and support vector machines (SVM) to build a passenger flow prediction model of rail transit in Beijing.

In recent studies, utilizing time series decomposition methods in hybrid models has become a new research direction to gain better prediction performance. The principle of time series decomposition methods is to decompose a complicated time series into multiple frequency components and predict them separately. Then, the predicted results are summed as the outcomes [12]. Commonly used time series decomposition methods include: empirical mode decomposition (EMD) [9,29], wavelet decomposition (WD) [30,31], singular spectrum analysis (SSA) [32,33], STL [34,35], etc. Chen et al. [29] proposed an EMD-based LSTM neural network model to predict short-term metro inbound passenger flow. Liu et al. [31] decomposed passenger flow data into high-frequency and low-frequency sequences through wavelet transform and the Mallat algorithm. The kernel extreme learning machine (KELM) approach was used to predict flows. Finally, different prediction sequences were reconstructed by wavelet transform. Qin et al. [34] utilized STL to decompose the monthly air passenger flow into three sub-series and adopted the improved echo state network (ESN) to predict each series. The prediction results of each series were summed to obtain the final prediction of monthly passenger flow. Chen et al. [35] used STL to decompose metro passenger flow data into three sub-series, and the LSTM neural network was employed to predict each decomposed series. All the predicted results were aggregated as the final result.

Until now, to our best knowledge, most studies have focused on predicting passenger flow under conventional conditions, and there are very few studies predicting passenger flow under the influence of COVID-19 or a similar pandemic. Thus, only a few studies related to COVID-19 and transportation are listed in this part: Liu et al. [36] proposed a method for re-establishing the passenger flow model and resetting the threshold value of alarm to predict and control uncertain passenger flow. Ku et al. [37] analyzed the changes in passengers’ travel behavior in Seoul due to the COVID-19 outbreak. They predicted that the demand would not fully recover to that in the period before COVID-19. Goenaga et al. [38] assessed the impact of closures related to the pandemic on traffic patterns for the state of North Carolina and the Commonwealth of Virginia. Chen et al. [39] established a dynamic programming model based on nonlinear integer programming to study the problem of boarding and alighting planning at various customized bus stops under the influence of COVID-19.

In conclusion, these studies suggest that hybrid models for short-term traffic passenger flow prediction can significantly improve accuracy. Thus, employing hybrid models based on time series decomposition is an effective way to improve prediction performance. However, it is still hard to provide high-precision prediction in emergencies, such as in the current case of the COVID-19 outbreak because of bus passenger flow having the characteristics of spatiotemporal complexity. Specifically, it has a highly complex relationship with these features: historical passenger flow, holidays, emergencies, weather conditions, etc. For instance, daily bus passenger flow is higher on weekdays and lower on weekends; when an epidemic or pandemic disease (like the current COVID-19 pandemic) occurs in an area, daily bus passenger flow decreases significantly as the number of confirmed cases rises. These irregular fluctuations can affect the accuracy of single model predictions. Therefore, analyzing these features and apply them to hybrid models is indispensable. Table 1 shows a comparison of some models mentioned above.

The remaining components of the research are organized as follows. The introduction of the proposed ISTL-LSTM model will be discussed in the forthcoming section. Section 4 describes and discusses the implementation of the proposed model in specific cases. The final section concludes the paper.

## 3. Methodology

In this paper, we proposed a new hybrid prediction model named ISTL-LSTM to solve the abovementioned problems. It can predict the short-term passenger flow during the COVID-19 pandemic or in conventional circumstances. The proposed model consists of nine steps: data cleaning, STL decomposition, data fusion, division, normalization, window sliding, prediction, de-normalization, and aggregation. Figure 1 demonstrates the flowchart of the proposed ISTL-LSTM model.

Data cleaning

Data quality significantly impacts the model’s prediction accuracy, so low-quality data should be cleaned before inputting to the model. Commonly used cleaning data methods include removing outliers, replacing missing values, smoothing noisy data, correcting inconsistent data, etc.

In this part, passenger flow data are processed to obtain passenger flow series Y. Feature data (e.g., weather data, holiday data, etc.) are processed to obtain n features $f_{n}$. In particular, Y and $f_{n}$ need to be processed using the same interval.

STL decomposition

First proposed by Cleveland in 1990 [40], STL is one of the robust and universal algorithms in time series decomposition. It has robust adaptation to outliers in the data and can be applied to a large number of time series data.

Given time series Y, STL could decompose $Y_{t}$ (the data of Y at timestep *t*) into three components: seasonality $S_{t}$, trend $T_{t}$, and residual $R_{t}.$
(1)$$Y_{t}=S_{t}+T_{t}+R_{t}\;\mathrm{for}\;t=1\;\mathrm{to}\;n$$

The specific decomposition process Y is as follows:

STL mainly consists of two loops, i.e., an inner loop and an outer loop. The inner loop is nested within the outer loop. The inner loop updates the seasonal and trend components with seasonal smoothing and trend smoothing. After each iteration of the inner loop, the residual component is calculated. The specific steps of the inner loop are shown as follows:

Suppose $S_{t}^{k}$ and $T_{t}^{k}$ are the seasonal and trend components at the kth iteration of the inner loop, and the initial value of $T_{t}$ is 0.

Step 1: Detrending. Subtract the trend component $T_{t}^{k}$ from the original value $Y_{t}$ to get a new $Y_{t}$.
(2)$$Y_{t}\leftarrow Y_{t}-T_{t}^{k}$$

Step 2: Cycle-Subseries Smoothing. Locally weighted regression (LOESS) [41] is used to smooth each sub-series from Step 1 and obtain the smoothing result $C_{t}^{k+1}$.

Step 3: Low-Pass Filtering. $C_{t}^{k+1}$ is handled by a filter that includes three moving average approaches. Then the result of low-pass filtering is handled by LOESS to obtain $L_{T}^{k+1}$.

Step 4: Detrending of Smoothed Cycle-Subseries. Get the seasonal components $S_{t}^{k+1}$.
(3)$$S_{t}^{k+1}=C_{t}^{k+1}-L_{t}^{k+1}$$

Step 5: De-Seasonalizing. Remove the seasonal component of $Y_{t}$.
(4)$$Y_{t}\leftarrow Y_{t}-S_{t}^{k+1}$$

Step 6: Trend Smoothing. Get the trend component $T_{t}^{k+1}$ by smoothing $Y_{t}$ using LOESS.

The residual component $R_{t}^{k+1}$ can be calculated when an iteration of the inner loop ends.
(5)$$R_{t}^{k+1}=Y_{t}-S_{t}^{k+1}-T_{t}^{k+1}$$

The primary function of the outer loop is to calculate the robustness weight $\rho_{t}$:(6)$$\rho_{t}=\frac{B\left(\left|R_{t}\right|\right)}{h}$$
(7)$$h=6\times median\left(\left|R_{t}\right|\right),$$
where B(x) is the bisquare function:(8)$$B\left(x\right)=\{\begin{array}{cc} {\left(1-x^{2}\right)}^{2} & \;\mathrm{for}\;0\leq x<1\\ 0 & \mathrm{for}\;x>1 \end{array}$$$\rho_{t}$ is used to down-weight the impact of outliers in Step 2 and Step 6 to reduce the influence of outliers on regression.

According to Equation (1), utilize STL to decompose $Y_{t}$ into three components by equal weights: seasonality $S_{t}$, trend $T_{t}$, and residual $R_{t}$. Then, obtain the corresponding sub-series: S, T, R.

Data fusion

For a stationary time series, good results can be obtained by direct prediction without using additional features. However, for non-stationary time series, it is difficult to get high accuracy when the prediction results are directly put into the model for training. Therefore, to improve the model’s prediction accuracy, it is necessary to introduce additional features.

In this part, the time interval is used as the associated field to concatenate each sub-series S, T, R with features $f_{n}$ to get new sub-series, S′, T′, R′.

Division

Before being put into the model, time series should be divided into training sets and test sets in a particular proportion.

Split S′, T′, R′ into a training set and a test set respectively to obtain $S_{a}^{\prime }$, $T_{a}^{\prime }$, $R_{a}^{\prime }$, and $S_{b}^{\prime }$, $T_{b}^{\prime }$, $R_{b}^{\prime }$, where index a is for the training set, and index b is for the test set. Due to the continuity of the time series, these series do not need shuffle processing when being split.

Normalization

Different features often have different units, which will affect the results of data analysis. In order to eliminate the influence of different units between indicators, data standardization is needed to solve the problem of comparability between data indicators.

Normalize the training sets $S_{a}^{\prime }$, $T_{a}^{\prime }$, $R_{a}^{\prime }$, and test sets $S_{b}^{\prime }$, $T_{b}^{\prime }$, $R_{b}^{\prime }$ separately. For example, $S_{a}^{\prime }\leftarrow normalize\;\left(S_{a}^{\prime }\right)$.

Window sliding

Before prediction, the original time series needs to be labeled. The window sliding algorithm is widely used in this case. Its principle is to use the data at timestep (t+1) as the label of data at timestep t, and create a corresponding set of labels.

In this part, set the window sliding timestep and prediction step in advance, and then convert the training set $S_{a}^{\prime }$, $T_{a}^{\prime }$, $R_{a}^{\prime }$, and test set $S_{b}^{\prime }$, $T_{b}^{\prime }$, $R_{b}^{\prime }$ into labeled datasets. For example, $S_{a}^{\prime }\leftarrow windowsliding\;\left(S_{a}^{\prime }\right)$.

Prediction

This study uses LSTM as the prediction model because it has good prediction performance for time series. LSTM was proposed by S. Hochreiter and J. Schmidhuber in 1997 [42]. It is a particular type of recurrent neural network (RNN) that is designed to avoid long-term dependency problems by remembering historical information.

LSTM adds three gates into each neuron of simple RNN: the input gate, the output gate, and the forget gate (Figure 2), all of which are controlled by the Sigmoid unit (0,1) [42].

The forget gate $f_{t}$ is used to control the historical information stored by the hidden layer node in the last time.
(9)$$f_{t}=\sigma \left(W_{f}\cdot \left[h_{t-1},x_{t}\right]+b_{f}\right),$$
where $f_{t}$ is the forget gate; σ is the sigmoid function; $W_{x}$ is the weight for the respective gate neurons; $h_{t-1}$ is the output of the previous LSTM cell; $x_{t}$ is the input; $b_{x}$ is the bias for the respective gate.

The input gate $i_{t}$ is used to control the current cell state.
(10)$$i_{t}=\sigma \left(W_{i}\cdot \left[h_{t-1},x_{t}\right]+b_{i}\right),$$
where $i_{t}$ is the input gate.

The output gate $o_{t}$ is used to control the output of the currently hidden layer node.
(11)$$o_{t}=\sigma \left(W_{o}\cdot \left[h_{t-1},x_{t}\right]+b_{o}\right),$$
where $o_{t}$ is the output gate.

The transition process from the original state $C_{t-1}$ to the new state $C_{t}$ is as follows:(12)$${\tilde{C}}_{t}=tanh\left(W_{C}\cdot \left[h_{t-1},x_{t}\right]+b_{C}\right),$$
(13)$$C_{t}=f_{t}\cdot C_{t-1}+i_{t}\cdot {\tilde{C}}_{t},$$
(14)$$h_{t}=o_{t}\cdot \tanh\left(C_{t}\right),$$
where $C_{t}$ is the cell state; ${\tilde{C}}_{t}$ is the candidate for cell state; tanh is the hyperbolic tangent function.

In the prediction part, three LSTM models are employed. The prediction process is as follows:

Step 1: Train three LSTM models with $S_{a}^{\prime }$, $T_{a}^{\prime }$, $R_{a}^{\prime }$ respectively.

Step 2: Test the trained models with $S_{b}^{\prime }$, $T_{b}^{\prime }$, $R_{b}^{\prime }$ to tune the best hyperparameters.

Step 3: Obtain three LSTM models $M_{S}$, $M_{T}$, $M_{R}$, and three best prediction results $S_{p}^{\prime }$, $T_{p}^{\prime }$, $R_{p}^{\prime }$.

De-normalization

In order to compare with the original series $Y_{t}$ and measure the performance of the model, the prediction results need to be de-normalized.

De-normalize the prediction results $S_{p}^{\prime }$, $T_{p}^{\prime }$, $R_{p}^{\prime }$ into readable data. For example, $S_{p}^{\prime }\leftarrow denormaling\;\left(S_{p}^{\prime }\right)$.

Aggregation

At the end of the ISTL-LSTM model, the prediction results need to be aggregated. Sum the prediction results $S_{p}^{\prime }$, $T_{p}^{\prime }$, $R_{p}^{\prime }$ according to equal weights into overall prediction results P, and evaluate.
(15)$$P=S_{p}^{\prime }+T_{p}^{\prime }+R_{p}^{\prime }$$

In short, the STL-LSTM prediction model is a hybrid model combining an STL model and three identical LSTM models (using the same hyperparameters) [35]. Based on the STL-LSTM model, the ISTL-LSTM prediction model introduces many related features of passenger flow and adopts different LSTM models (hyperparameters can be different). In other words, the ISTL-LSTM prediction model is a hybrid model combining an STL model, many related features of passenger flow, and three LSTM models.

## 4. Case Study

In the case study, the prediction of Beijing bus passenger flow under the influence of the COVID-19 pandemic was conducted as a research project. To evaluate the prediction performance of the proposed ISTL-LSTM model, these models were added for comparison: STL-LSTM [35], LSTM, gated recurrent unit (GRU), linear regression (LR), k-nearest neighbor regression (KNR), and extreme gradient boosting (XGBoost).

This section has been divided into six parts. The first part deals with research data gathering and description in detail. Part two begins by preprocessing the research data before modeling. Part three is concerned with utilizing the robust decomposition techniques and focuses on how to integrate the data. Section four employs the hyperparameter tuning techniques for the proposed models. The fifth part uses well-known calculation metrics for evaluation of the models. The remaining part of the case study presents the findings and results of the prediction models and gives an analysis and further comparison.

### 4.1. Data Description

Data sources of this research are in four parts: the daily passenger flow of the bus system in Beijing, holidays and weekends, weather conditions, and the daily report of COVID-19 cases in Beijing. The frequency of data collection was daily. The comprehensive descriptions of these original datasets are explained in Table 2.

Bus daily passenger flow data were sourced from the bus daily passenger flow report (in Excel format) provided by the Transportation Operations Coordination Center (TOCC), covering the whole region of Beijing. As illustrated in Figure 3, the trend of passenger flow is represented with the blue curve, compared with the number of newly confirmed cases of COVID-19 in Beijing with the gray curve between November 2019 and October 2020.

Holiday data of Chinese statutory public holidays were observed from 2019 and 2020 holiday schedules [43,44] issued by the General Office of the State Council of the People’s Republic of China.

Weather data and corresponding weather conditions were gathered from Nanyuan monitoring points in Beijing, downloaded from Wunderground [45]. The main fields included temperature, precipitation, and so forth. 

COVID-19 data were drawn from GitHub [46]. They were crawled by a web scraper on GitHub [47] from a real-time COVID-19 website named DXY [48]. This website aggregates comprehensive and official statistics of COVID-19, including daily cumulative confirmed cases, cumulative cured cases, cumulative deaths, newly confirmed cases, existing confirmed cases, and newly cured cases around the globe. COVID-19 data of Beijing (Figure 4) were then filtered from the collected data and compared with the official statistics of COVID-19 published by the Beijing Municipal Health Commission (BMHC) [49], which formed the final dataset of COVID-19 in Beijing. Figure 3 represents the overall trend of passenger flow and the number of newly confirmed cases.

As is illustrated in Figure 3, the number of newly confirmed cases in Beijing continued to rise after the first confirmed case appeared in January. During the time from January to April, new cases were confirmed every day. It can be seen that the number of newly confirmed cases fluctuated considerably during these periods, reaching two peaks at approximately 29 people per day. In April 2020, newly confirmed cases declined sharply to nearly 0, and this circumstance was maintained for more than one month. Subsequently, the pandemic broke out again, causing the number of newly confirmed cases to reach the highest peak in history, with nearly 40 people per day, and then fall dramatically to 0 cases over nearly one month. To sum up, there were two stages of the newly confirmed cases that fluctuated dramatically in 2020. The first stage was from January 2020 to April 2020, while the second stage was approximately from June 2020 to July 2020.

COVID-19 data are like a barometer of bus passenger flow and are closely related to it. After the first new case appeared in January 2020, the daily bus passenger flow in Beijing declined sharply and reached the lowest point, with no passengers taking the bus. In March 2020, the pandemic was still robust, and the newly confirmed cases reached their secondary peak. Since pandemic prevention measures were continuously being upgraded, some residents began to take buses under effective safety measures. Therefore, daily passenger flow increased slowly and gradually and with no apparent fluctuation during the span. In turn, with the new confirmed cases having been cleared from April to May in 2020, the volume of bus passenger flow increased gradually until the secondary outbreak of the pandemic. Subsequently, although the pandemic harmed bus passenger flow in June, with the number of passengers decreasing by approximately 4,000,000, the impact was not as significant as that during the first outbreak in the first phase (from January 2020 to April 2020). This scenario implies that practical and comprehensive prevention measures for Beijing citizens and the timely and highly efficient governing capacity motivated the residents’ confidence in traveling when choosing public transportation, so, passengers no longer completely gave up public transportation as in the early days of the outbreak.

### 4.2. Data Preprocessing

In order to improve the prediction models’ accuracy, each original dataset needs to be preprocessed. In this case study, considering utilizing a range of models in the research, data preprocessing is divided into two categories: preprocessing for decomposed models (such as ISTL-LSTM) and preprocessing for non-decomposed models (such as LR, KNR, etc.). The preprocessing of non-decomposed models consists of two components. The first step is to clean the data, while the second step is to integrate the cleaned data with the features, such as holiday, weather, and COVID-19. Preprocessing of the decomposition model is explained in Section 4.3, consisting of three components. The first step is to clean the data, and the second step is to decompose the cleaned data. Furthermore, the third step is to integrate the decomposed data with features.

The passenger flow dataset is a statistical dataset obtained from TOCC, an official and authoritative agency in China, which means data quality is relatively higher. Therefore, the data could be used directly without preprocessing. In terms of the holiday dataset, it is easier to identify weekdays and holidays in the calendar. Hence, labeling the weekdays and transforming them with a one-hot encoding technique were required for implementation. As for the weather dataset gathered from the website, the quality of the data was not ideal. For instance, some weather information was missing for some days, so it was necessary to find other sources to fill the missing values. The COVID-19 dataset was extracted from websites using web crawlers. Since the dataset covered global COVID-19 information, COVID-19 data in Beijing first had to be filtered and verified with the daily COVID-19 report issued by BMHC.

After data cleaning was completed, the passenger flow dataset, the holiday dataset, the weather dataset, and the COVID-19 dataset were merged into one dataset through the public field ‘DATE’. After normalization, the fused dataset could be directly utilized in some non-decomposed models compared to the proposed ISTL-LSTM model.

### 4.3. STL Decomposition and Refusion

For the proposed ISTL-LSTM model in this research, without using the integrated data, it was necessary to decompose the cleaned passenger flows firstly and then integrate them with the holiday, weather, and COVID-19 features.

First, the STL method decomposed the passenger flow series into trend series, seasonal series, and residual series by equal weights, and the results are shown in Figure 5. The sum of the three sub-series equaled the original passenger flow series, and there was no error during this process. The trend series describes the overall trend of bus passenger flow, with the seasonal series presenting the cyclical changes of bus passenger flow and the residual series describing other items.

Second, after the decomposition process was completed, the sub-series was regarded as a new time series of passenger flow. The holiday dataset, the weather dataset, and the COVID-19 dataset were then integrated with each new sub-series through the field ‘DATE’, to obtain three new datasets: a fused trend dataset, a fused seasonal dataset, and a fused residual dataset.

### 4.4. Experimental Setup

For model development, the fused datasets from Section 4.2 and Section 4.3 were divided, with a ratio of 70%:30% being used in training and testing stages, respectively. Then, the training sets and test sets were normalized by using the training sets as the standard. The sliding window algorithm was employed to add labels for the time series. In this research, the hyperparameter—‘look back’ was set to 1. All models were set with the same seed 1000 to reproduce the experimental results.

Based on an enormous amount of experiment results, hyperparameters of models (ISTL-LSTM, STL-LSTM, LSTM, and GRU models) were set as follows: the number of hidden layers was two; the number of neurons in each layer was 100; the range of training epochs was 50 to 150 with the step size 5 to identify the best epochs for each model; then batch size was set to 400 to cover entire timesteps; neurons were dropped from the network with the dropout rate of 0.2; the activation function was tanh; the loss function was mean squared error (MSE) and the optimizer was adaptive moment estimation (Adam). The hyperparameters mentioned above are listed in Table 3.

It can be seen from Table 4 that the listed hyperparameters are from KNR and XGBoost models. In order to reach the highest scores, the hyperparameters were tested by 10-fold grid search, utilizing MSE as a scoring metric.

Due to the previous normalization, the results of the trained model cannot be understood directly. In this case, the predicted values were required to be de-normalized, which is more readable and comparable with the actual value.

### 4.5. Evaluation Metrics

Evaluation metrics enable the evaluation of the performance of the tested models. In the research, evaluation metrics were composed of four methods, including the mean absolute error (MAE), the mean absolute percentage error (MAPE), the root mean square error (RMSE), and the variance absolute percentage error (VAPE). Each mathematical function of the method is defined as below:(16)$$\mathrm{MAE}=\frac{1}{n}\overset{n}{\underset{i=1}{\sum }}\left|y_{i}-{\hat{y}}_{i}\right|$$
(17)$$\mathrm{MAPE}\left(\%\right)=\frac{1}{n}\overset{n}{\underset{i=1}{\sum }}|\frac{y_{i}-{\hat{y}}_{i}}{y_{i}}|\times 100\%,$$
(18)$$\mathrm{RMSE}=\sqrt{\frac{1}{n}\overset{n}{\underset{i=1}{\sum }}{\left(y_{i}-{\hat{y}}_{i}\right)}^{2}}$$
(19)$$\mathrm{VAPE}\%=\frac{1}{n}\overset{n}{\underset{i=1}{\sum }}{\left(\frac{y_{i}-{\hat{y}}_{i}}{y_{i}}\right)}^{2}\times 100\%,$$
where $y_{i}$ is the actual value, which is the label of the data; ${\hat{y}}_{i}$ is the predicted value; n is the number of the predicted value. The higher the MAE, MAPE, RMSE, and VAPE values, the lower the accuracy of prediction results. In contrast, the lower the values, the higher the accuracy.

### 4.6. Results and Discussion

As shown in Figure 6, the prediction performance of various deep neural network models (ISTL-LSTM, STL-LSTM, LSTM, GRU) can be compared. The results indicate that when the hyperparameter ‘epochs’ was equal to 65, 100, 65, or 50, respectively, the performance of the corresponding model could be most optimal.

Figure 7 compares the results of the two hybrid models—the ISTL-LSTM model and the STL-LSTM model, respectively. As is shown in Figure 7, a fluctuation among actual values, predicted values of the training set, and predicted value of the test set is illustrated. 

Figure 7a represents the prediction results of the ISTL-LSTM model. Considering several features fed into the model, such as data for holidays, weather conditions, and circumstances of COVID-19, prediction values fluctuated significantly, which is demonstrated clearly in the (orange) curve of the predicted value of the training set. Predicted values of the test set were particularly close to the actual values, which means that the prediction result errors (bias and variance) were few. Therefore, the model performed well during the prediction. 

Figure 7b demonstrates the prediction result of the STL-LSTM model, including actual values, predicted values of the training set, and test set. The prediction performance of the STL-LSTM model was different from that of the ISTL-LSTM model, as shown in Figure 7a. In other words, predicted values of the training set in the STL-LSTM model decreased more sharply during the intervals from the 75th day to the 100th day approximately. The reason is that the STL-LSTM model only uses the previous day’s passenger flow to predict the next day’s flow instead of using various features (weather conditions, holidays, and the circumstances of COVID-19). Besides, in the STL-LSTM model, the curve shape of the predicted values of the training and test sets could be regarded as implementing a horizontal remove at 1 unit and a flattened transformation, compared with the actual value curve. In this scenario, the ISTL-LSTM model utilizing various features enables us to make more realistic predictions. In addition, it can be seen from Figure 7b that the variance of the prediction results of the test set was relatively high.

Figure 8 compares the results of the two single models of neural networks—LSTM and GRU, respectively. As mentioned before, the prediction accuracy of a single model is generally lower than that of a hybrid model. 

Therefore, it can be seen from Figure 8a that the variance of the prediction results of the test set was relatively high compared with that of the ISTL-LSTM model, which means the prediction accuracy of LSTM was inferior to that of the ISTL-LSTM model. This point also proves the thesis that hybrid models are usually superior.

Figure 8b represents the prediction results of the GRU single model. Due to it being a single model with relatively lower accuracy compared with hybrid models, the GRU model had more errors. Therefore it is not recommended to predict passenger flow in emergency cases. On the other hand, both LSTM and GRU models input various features, such as data on weather conditions, holidays, and the circumstances of COVID-19. Therefore their prediction curves are similar to that of the ISTL-LSTM model.

As for the prediction performance of LR, KNR, and XGBoost, those models performed inefficiently. In the research, their result figures were not displayed. Their variances were very high, so they are not applicable for accurate prediction either. It is also proved that the prediction accuracy of parametric models and non-parametric models is generally low.

We used four metrics (Section 4.5)—MAE, RMSE, VAPE, and MAPE—to evaluate the accuracy of the prediction. The comparison results are shown in Table 5.

It can be seen from Table 5 that under the impact of COVID-19, the prediction performance of the GRU model measured poorly by each metric. The LR, KNR, and XGBoost models performed relatively better, but their prediction performance was still not as good as that of the LSTM and hybrid models (STL-LSTM and ISTL-LSTM). Therefore, it can be concluded that hybrid models have a more substantial prediction capacity compared with a single model.

When comparing STL-LSTM and ISTL-LSTM hybrid models, it can be noted that the prediction results of the ISTL-LSTM model were much better than those of the STL-LSTM model on all metrics with the prediction accuracy being improved by at least 15%. The reason is that the ISTL-LSTM model inputs new features on the base of the STL-LSTM model and uses different hyperparameters for LSTM models.

An interesting fact is that the results show the LSTM model performed better than the STL-LSTM model. This implies that some single models with features may have higher accuracy than hybrid models without any features in emergency cases, such as the current COVID-19 pandemic, which proves the importance of the features in improving the accuracy.

## 5. Conclusions

The main goal of this study was to design a hybrid model named ISTL-LSTM for passenger flow prediction. In the case study, we used the data during the COVID-19 pandemic in Beijing to test the performance of the proposed model. This research supports the idea that compared with some single models and the STL-LSTM hybrid model, the ISTL-LSTM model can effectively predict daily bus passenger flow. The accuracy can be improved by at least 15%.

The proposed ISTL-LSTM model can be applied to bus passenger flow prediction after the COVID-19 outbreak to monitor the trend of daily bus passenger flows under the impact of the pandemic. Based on the predicted passenger flow, the public transport companies, such as TOCC and BPTC, can make better decisions on operation scheduling, including controlling the time interval between bus departures, and adjusting the number of buses and staff to effectively control operating costs. In addition, predicted passenger flow is also important to epidemic prevention and early warnings to help these companies make quick responses to an epidemic or pandemic disease in advance, such as controlling the maximum number of passengers on each bus and the minimum passenger distance.

The next step of our research is to add other related features from the bus index system to the proposed model and employ integrated models for sub-series prediction based on this research. This will further improve the accuracy of model prediction and the robustness of the entire model.

## Figures and Tables

**Figure 1 sensors-21-05950-f001:**
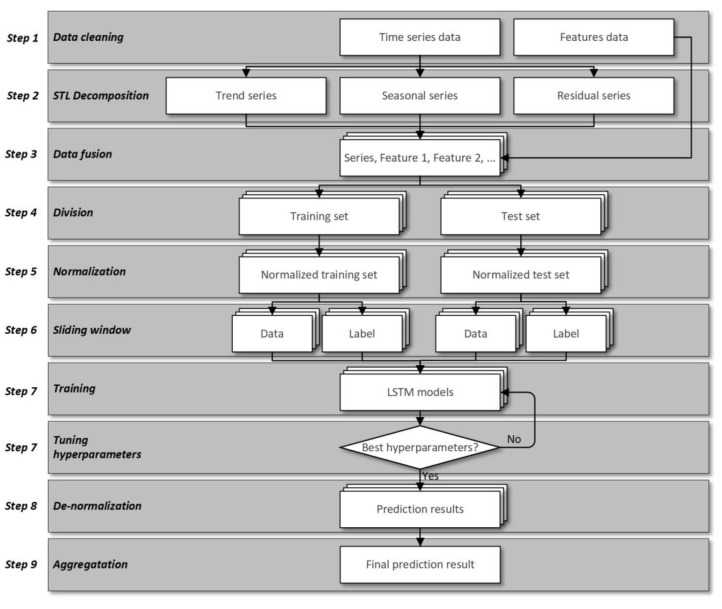
Flowchart of ISTL-LSTM.

**Figure 2 sensors-21-05950-f002:**
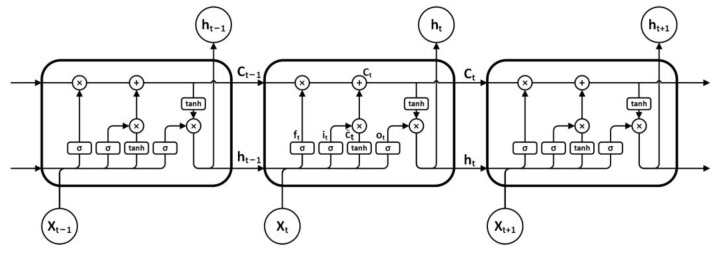
Illustration of LSTM.

**Figure 3 sensors-21-05950-f003:**
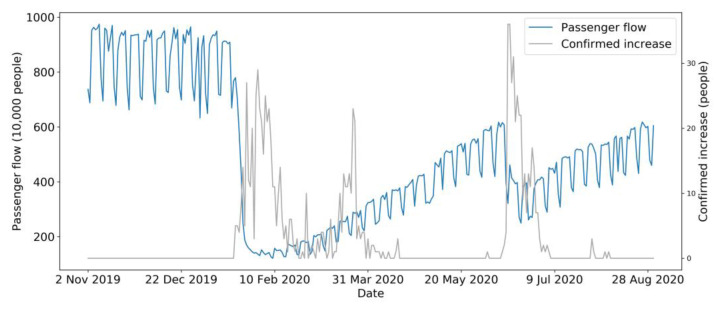
Curves of bus passenger flow and COVID-19 newly confirmed cases.

**Figure 4 sensors-21-05950-f004:**
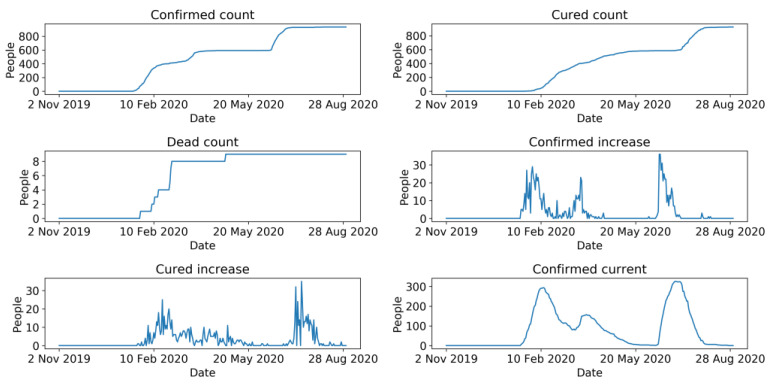
Curves of COVID-19 dataset.

**Figure 5 sensors-21-05950-f005:**
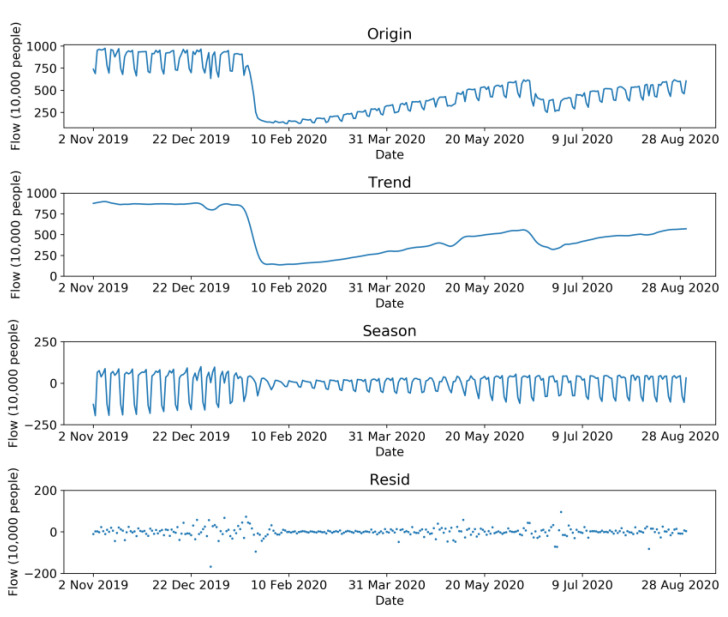
STL decomposition results of bus passenger flow.

**Figure 6 sensors-21-05950-f006:**
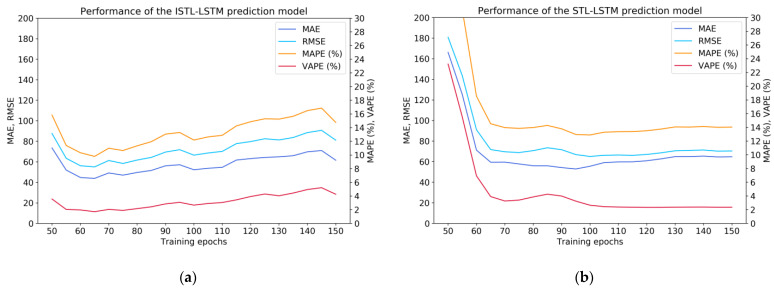
Prediction performance of four deep neural network models: (**a**) performance of the ISTL-LSTM model (the performance was the best when epochs = 65); (**b**) performance of the STL-LSTM model (the performance was the best when epochs = 100); (**c**) performance of the LSTM model (the performance was the best when epochs = 65); (**d**) performance of the GRU model (the performance was the best when epochs = 50).

**Figure 7 sensors-21-05950-f007:**
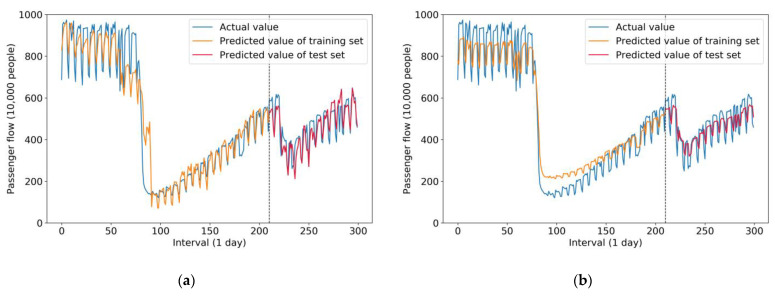
Comparison results of the predicted value and the actual value in two hybrid models: (**a**) comparison results of the ISTL-LSTM model; (**b**) comparison results of the STL-LSTM model.

**Figure 8 sensors-21-05950-f008:**
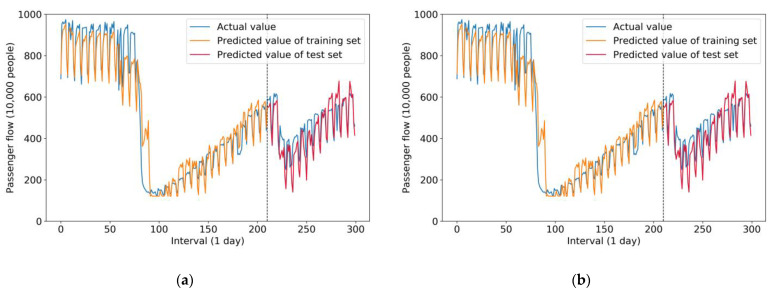
(**a**) Prediction curve of the LSTM model with features; (**b**) prediction curve of the GRU model with features.

**Table 1 sensors-21-05950-t001:** Comparison of some existing methods concerning passenger flow prediction.

Categories	References	Models	Scenarios	Applicable for Emergencies	Accuracy
Parametric models	Milenkovic et al. [14]	SARIMA	Railway	Bad	Low
Non-parametric models	Tang et al. [2]	SVR	Urban rail transit	Bad	Medium
Deep learning methods	Bai et al. [25]	DBN	Bus	Bad	Low
Deep learning methods	Liu et al. [4]	LSTM	Urban rail transit	Bad	Medium
Deep learning methods	Liu et al. [13]	SAE	Bus	Bad	Medium
Hybrid models	Sun et al. [30]	Wavelet-SVM	Urban rail transit	Medium	Medium
Hybrid models	Shang et al. [32]	SSA-KELM	Expressway	Medium	High
Hybrid models	Chen et al. [29]	EMD-LSTM	Urban rail transit	Medium	High
Hybrid models	Chen et al. [35]	STL-LSTM	Urban rail transit	Medium	High
Hybrid models	The proposed model	ISTL-LSTM	Bus	Good	High

**Table 2 sensors-21-05950-t002:** Data sources.

Datasets	Data Description	Number of Records	Data Size
Daily passenger flow dataset	Including the daily number of bus passengers during the pandemic from 2 November 2019 to 31 August 2020 in Beijing.	305	11 KB
Holiday dataset	Holiday information for 2019 and 2020 in Beijing, including 1 field: whether the day is a holiday.	731	12 KB
Weather dataset	Weather information for 2019 and 2020 in Beijing, including 6 fields: temperature, dew point, humidity, wind speed, pressure, and precipitation.	731	82 KB
COVID-19 dataset	COVID-19 information from 22 January 2020 to 21 January 2021, including suspected count, cured count, death count, confirmed count, etc.	427,508	44.5 MB

**Table 3 sensors-21-05950-t003:** Hyperparameters of deep neural network models (ISTL-LSTM, STL-LSTM, LSTM, GRU).

No.	Hyperparameters	Values
1	Units of hidden layer 1	100
2	Units of hidden layer 2	100
3	Epochs	{50,55,60,65,70,75,80,85,90,95,100,105,110,115,120,125,130,135,140,145,150}
4	Batch size	400
5	Dropout	0.2
6	Activation function	tanh
7	Loss function	MSE
8	Optimizer	Adam

**Table 4 sensors-21-05950-t004:** Hyperparameters of KNR and XGBoost models.

No.	Models	Hyperparameters	Values
1	KNR	Weights	Distance
2	KNR	Leaf size	30
3	KNR	*p*	1
4	KNR	Neighbors	6
5	XGBoost	Learning rate	0.01
6	XGBoost	Max depth	6
7	XGBoost	Estimators	500

**Table 5 sensors-21-05950-t005:** Performance comparison of the proposed model and other models. Bold means that these figures are the smallest in each column respectively, which means the best performance.

Model	MAE	MAPE (%)	RMSE	VAPE (%)
ISTL-LSTM	**43.81**	**9.79**	**55.03**	**1.70**
STL-LSTM	55.50	12.88	65.01	2.65
LSTM	52.20	12.29	66.17	2.70
GRU	71.47	17.11	87.29	5.04
LR	63.27	14.15	78.43	3.05
KNR	65.57	15.15	77.81	4.11
XGBoost	63.21	16.41	86.32	6.45

## Data Availability

Bus daily passenger flow data can be obtained from the corresponding author. Holiday data can be obtained at http://www.gov.cn/gongbao/content/2018/content_5350046.htm (accessed on 3 August 2021) and http://www.gov.cn/zhengce/content/2019-11/21/content_5454164.htm?trs=1 (accessed on 3 August 2021). Weather data can be obtained at https://www.wunderground.com/history/ (accessed on 1 July 2021). COVID-19 data can be obtained at https://github.com/BlankerL/DXY-COVID-19-Data/releases/tag/2021.08.03 (accessed on 3 August 2021).

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
