# Peer review of "An Improved STL-LSTM Model for Daily Bus Passenger Flow Prediction during the COVID-19 Pandemic"

_sensors, 2021, doi:10.3390/s21175950_

Round 1

Reviewer 1 Report

This work presents a prediction scheme called the improved ISTL-LSTM model which combines seasonal-trend decomposition procedure based on loess, with weather, COVID-19, and holiday data, and long short-term memory neural network. The experimental results confirm the suitability of the proposed model to predict the bus daily passenger flow for the Beijing Public Transportation Corporation operation scheduling.

The results are significant for the knowledge field and the conclusions are supported by the experiment results. Nevertheless, some modifications are necessary before the manuscript could be considered for publication.

The figures (Figure 1, Figure 2, and Figures 4 to13) are too small and should be enlarged to improve readability.

The link provided in lines 267-268, should be only cited in the text, and then included in the References section “Covid-19 from the Beijing Municipal Health Commission (BMHC, http://wjw.bei-jing.gov.cn/wjwh/ztzl/xxgzbd/gzbdyqtb/)”

Conclusion section should provide quantitative results, instead of qualitative comments “The experiment indicates that the ISTL-LSTM model can improve the accuracy of prediction”

Reviewer 2 Report

The Abstract of the manuscript-at-hand needs to be considerably improved primarily in terms of (a) bringing in the underlying rationale (i.e., motivation) of undertaking this particular study and (b) significant contributions made to the existing body of literature. Also, there are a number of CoVID-19 related studies already published in the research literature (i.e., within the past year or so) and a reference to some of them is highly indispensable here.

Figure 3 (The Flowchart of the Proposed Improved STL-LSTM Model) needs to be elaborated in a more succinct and categorical manner. Also, it is not clear to a reader how the data employed in the manuscript-at-hand was really crawled and what it took to aggregate the same from disparate sources. Furthermore, how the hyperparameters were set, i.e., the authors stated that they were set as a result of many experiments but have not been categorical on the same. Critical analysis in Section 3.4 is also missing, i.e., Figure 8 - 13 needs to be elaborated in a more logical manner as merely describing them does not do any justice here.

There are considerable issues in terms of jargon and sentence structure and major proofreading is required in this regard. Also, the authors need to address the structure of the manuscript-at-hand, i.e., Parametric Models (on Line 62), Nonparametric Models (on Line 67), Deep-learning Models (on line 75), and Hybrid Models (on Line 91) all appear to be sub-headings and should be arranged as such.

Finally, some old literature has been cited in the manuscript-at-hand (in the References section), e.g., from the years 2002, 2009, and 2010, and it would be appreciated if the same could be replaced with the state-of-the-art. 

Reviewer 3 Report

### General considerations

The topic discussed in the paper is interesting but the use of the English language is poor and the organization of the paper is weak.
The novelties are somehow limited to the material reported in Section 2.3.
Indeed, Sections 2.1 and 2.2, part of the core part, look more like background than part of the contributions or, at least, the Authors fail to explain if there is any novelty in these sections.

The experimental section is quite long, and the final results are promising and interesting.
Nevertheless, experiments are also full of useless tables and questionable graphics.
For sure, as it stands now the experimental part needs strong adjustments to report only meaningful results and to describe them properly.

### Organization

The organization of the paper needs drastic improvements.

The "Introduction" section describes the methodology in a few lines (128-138).
Moreover, this description is very similar to the one reported in the abstract (lines 19-29) and it does not add any details or information.
This is far to be sufficient to describe the work.
Indeed, the introduction section is full of comments on "related works" (lines 51-127) whereas I think that these comments should be moved forward to a separate "Related works" section.

The authors should better clarify how to use their flow prediction from a practical point of view as, from their current explanation, the only outcome seems to be "increasing or decreasing the bus frequency".

The authors also use "embedded" paragraph titles on many occasions.
See for examples lines 62, 68, 75, and 91, then again lines 193, 197, and 200.
These "embedded" titles are confusing and difficult to spot and they should be transformed into items or proper separate paragraphs.

I wonder whether Figure 1 and Figure are really useful.
Can authors insert some explanation/comments on them, if they are?

Page 6 is partially empty.
Tables from 2 to 4 are useless.
Figures are too many, too small, and include pretty unreadable captions.

### English style

The English style is poor and the paper needs a main rewriting stage.
Here there is a partial list of problems:
- The text presents words repetitions in many phrases: "improved" appears twice in the same phrase in line 19; "process" is present twice in line 21; "forecasting" appears 4 times in lines 49-51 and 3 times in lines 116-118; etc.
- Line 23: "Then ..." -> "Then, "
- Line 28:  Please, move the acronym BPTC and its explanation to line 44
- Line 41:  "as an example, the outbreak" ->  "as an example. The outbreak"
- Lines 115-126:  "In conclusion, ..." and "However" should be inserted in the same paragraph. Please, insert a new line before "In conclusion" and remove it before "However".
- Line 122: "When ... will decrease ..." -> "When ... decreases ..."
- Line 125:  "the above hybrid model" -> which one? You discuss several hybrid models.
- Line 208: There is a certain confusion between the use of the acronym STL-LSTM and ISTL-LSTM in the text.
- Figure 3:  I would modify almost all "captions", as: "STL Decomposing" -> "STL Decomposition",  "Dividing" -> "Division", etc., "Aggregating" -> "Aggregation"
- Line 242: "... at last.", please rephrase.
- Table 1: "The number of records" -> "Number of records"
- Line 275: "and continues to rise" -> Hug? What does continue to rise? The "first confirmed case"? Please, rephrase.
- Line 300: "accuracy of models" -> which models?
- Line 371: "results and discussion" -> "Results and discussion"
- Line 372: "Figure 8 is the ..." -> "Figure 8 reports ..." ?
- Caption of Figure 8 (and all following figures, essentially): "... with factors" -> Hug? Unclear. Please, explain.
- Many phrases are unnecessarily long and cumbersome, please split them and rephrase. Examples of these phrases are in lines 423-427, 434-439, etc.

### Experimental results

All figures (from Figure 4 on) are too small and they also include tiny captions and labels that are almost unreadable.
Tables 2, 3, and 4 are not particularly useful nor interesting.
The explanation reported in the text is far more than sufficient to describe the process.

The core results are described in Figures 8-12.
I wonder whether the prediction on the trend, seasonal, and residual series are really meaningful (in all cases) or only the final prediction (plot (a) in all Figures) would be sufficient.

Reviewer 4 Report

This paper deals with the problem of predicting passenger flows for a public transport company in cases of severe changes in public transport user’s behavior. A new method based on local regression and long-short term memory neural networks is proposed. The paper deals with an actual problem of daily passenger flow using machine learning approaches. The newly proposed method has some potential. For a better argumentation, the paper writing must be improved, and the obtained error results have to be discussed in more detail. Emphasize the advantages of your proposed method in comparison to existing approaches.

Thus, to improve your paper, pay attention to the following:

  • Explain all of your abbreviations when first used. The abstract is an independent text and treat it so in all aspects (abbreviations and paper overview);
  • Mention your use case for testing in the abstract and the general applicability of your proposed method. Currently, the abstract reads that your method is made only for one company, and this is a poor result considering that your method should work will other similar data sets as well;
  • Do not emphasize solving your local passenger forecasting problem so much. It’s a global problem, and you used your local data for forecasting. New York, Berlin, London, Paris, Singapore, Shanghai, … have the same problem and would gladly use your method. Your method solves the global problem, and you test it using your available local data;
  • Explicitly state your scientific contributions in the Introduction section;
  • Add punctuation to equations being part of a sentence;
  • Enlarge Figs. 1, 2, 4, 5, 6, 7, 8, etc. or enlarge the font to make it more readable;
  • You mention (un)supervised time series. What do you mean by this? Data sets or approaches for (un)supervised learning of your data?
  • Word data is plural;
  • Table 5 contains very large MAPE values. Do you want to use a method that makes a prediction with more than 5 or 10% average error? Carefully check your data and comment on this in more detail. I suppose your forecasts are too high.

There are some other comments in the attached PDF.

Round 2

Reviewer 2 Report

Thank you for carrying out a comprehensive revision of the manuscript-at-hand. I am pleased with the changes made in the same. 

In a last bid to improve the quality of the content, kindly do a quick check on the language to resolve any leftover issues, i.e., for instance, Line 50 states, "Until now, to our best knowledge, most studies focus on predict passenger flow ...', i.e., most studies focus on predicting passenger, and so on.

Author Response

Point 1: In a last bid to improve the quality of the content, kindly do a quick check on the language to resolve any leftover issues, i.e., for instance, Line 50 states, "Until now, to our best knowledge, most studies focus on predict passenger flow ...', i.e., most studies focus on predicting passenger, and so on.

Response 1: Thanks for your suggestion. I have carefully checked all the leftover issues and improved this paper with a language polishing service.

Reviewer 3 Report

The authors made substantial changes to their original work and the new version is definitely more readable than the original one.
Unfortunately, novelties are still somehow limited.
Moreover, and most importantly at this stage,  the use of the English language is still sub-standard.
Errors, typos, and inelegant phrases are too many to mention.
Here there is a couple of examples just in the first half of a page:

"The Covid-19 pandemic-a major public health problem-confronting the globe is enormous, which causes difficulty and trouble for both people’s travel and public transport companies’ 12 management."

"In order to identify the process of the ISTL-LSTM model, it is divided into three parts."

I think that to be acceptable, the authors should resort to a professional proofreader to revise the entire manuscript.

As already stated before, the final results are promising but the quality of the experimental section should be improved.
As before, my suggestion is to reduce the "number" of graphics presented and, at the same time, to improve the quality and quantity of the comments and considerations reported for the remaining ones.

Reviewer 4 Report

The paper has been significantly improved. All of my comments are taken into account.

There are some minor comments in the attached PDF.

Author Response

Point 1: There are some minor comments in the attached PDF.

Response 1: Thanks for your suggestions. The manuscript has been revised according to your suggestions. Please see the new revision.